# Propolis: A Detailed Insight of Its Anticancer Molecular Mechanisms

**DOI:** 10.3390/ph16030450

**Published:** 2023-03-16

**Authors:** Suhib Altabbal, Khawla Athamnah, Aaesha Rahma, Adil Farooq Wali, Ali H. Eid, Rabah Iratni, Yusra Al Dhaheri

**Affiliations:** 1Department of Biology, College of Science, United Arab Emirates University, Al-Ain P.O. Box 15551, United Arab Emirates; 201970235@uaeu.ac.ae (S.A.);; 2Department of Biology, College of Arts and Sciences, Khalifa University, Abu Dhabi P.O. Box 127788, United Arab Emirates; 3Department of Pharmaceutical Chemistry, RAK College of Pharmacy, RAK Medical and Health Sciences University, Ras Al Khaimah P.O. Box 11172, United Arab Emirates; 4Department of Basic Medical Sciences, College of Medicine, QU Health, Qatar University, Doha P.O. Box 2713, Qatar

**Keywords:** propolis, biological activities, cancer, molecular mechanisms, natural products

## Abstract

Cancer is the second most life-threatening disease and has become a global health and economic problem worldwide. Due to the multifactorial nature of cancer, its pathophysiology is not completely understood so far, which makes it hard to treat. The current therapeutic strategies for cancer lack the efficacy due to the emergence of drug resistance and the toxic side effects associated with the treatment. Therefore, the search for more efficient and less toxic cancer treatment strategies is still at the forefront of current research. Propolis is a mixture of resinous compounds containing beeswax and partially digested exudates from plants leaves and buds. Its chemical composition varies widely depending on the bee species, geographic location, plant species, and weather conditions. Since ancient times, propolis has been used in many conditions and aliments for its healing properties. Propolis has well-known therapeutic actions including antioxidative, antimicrobial, anti-inflammatory, and anticancer properties. In recent years, extensive in vitro and in vivo studies have suggested that propolis possesses properties against several types of cancers. The present review highlights the recent progress made on the molecular targets and signaling pathways involved in the anticancer activities of propolis. Propolis exerts anticancer effects primarily by inhibiting cancer cell proliferation, inducing apoptosis through regulating various signaling pathways and arresting the tumor cell cycle, inducing autophagy, epigenetic modulations, and further inhibiting the invasion and metastasis of tumors. Propolis targets numerous signaling pathways associated with cancer therapy, including pathways mediated by p53, β-catenin, ERK1/2, MAPK, and NF-κB. Possible synergistic actions of a combination therapy of propolis with existing chemotherapies are also discussed in this review. Overall, propolis, by acting on diverse mechanisms simultaneously, can be considered to be a promising, multi-targeting, multi-pathways anticancer agent for the treatment of various types of cancers.

## 1. Introduction

Cancer is a disease that is severely affecting the world population [1]. It has been estimated that by 2030, cancer will increase to 26 million cases yearly in the whole world. Although many strategies and drugs have been developed to treat or prevent cancer, it is still a global concern [2,3]. Currently, there are different options for cancer treatment including surgery, radiotherapy, and chemotherapy [4,5]. Although removing the tumor by surgery is an effective way to treat cancer, it is not easy to detect cancer at an early stage. Treatments like chemotherapy and radiography have side effects, can damage normal cells, and cancer cells may develop drug resistance that leads to a lack of response to chemotherapeutic agents, which hinder their clinical uses. These adverse effects reduce the quality of life in the patients and might be a reason to stop the therapy. In addition, current cancer treatments are expensive; therefore, the search for alternatives is urgently required [6,7,8,9]. Recently, many studies have reported various natural alternative therapies for the treatment of cancer. Different studies showed that drugs derived from natural products might be an alternative because of their availability, higher effectiveness, and fewer side effects. In addition, there are numerous natural products that have been found to target multiple cancer signaling pathways simultaneously [6,10,11]. Propolis (bee glue) is a collection of resinous compounds containing beeswax and partially digested exudates from plants leaves and buds. It is produced and used by bees as a source of protection from predators and as a stabilizing seal for the honeycomb structure. Its color, like its composition, varies widely from green to brown, yellow, and sometimes red. Depending on the bee species, geographic location, plant species, and weather conditions, the chemical makeup of propolis fluctuates significantly [12]. As the domestic applications of honey expand in multiple communities, the uses and properties of propolis have become well understood. Studies have recently indicated that propolis has a variety of pharmacological activities including, antibacterial effects, antioxidative, fungicidal, antiparasitic, and anti-inflammatory properties [13]. Propolis has been reported to have cytotoxic effects on numerous cancer cell lines [14,15]. It has been demonstrated that propolis mediates not only the intrinsic (mainly mitochondrial) but also the extrinsic apoptotic cancer cell death [16,17,18]. In addition to apoptosis, propolis plays an important role in arresting the cell cycle via controlling the expression of CDK enzymes (cyclin-dependent kinases) [19,20,21,22]. Propolis has also been shown to suppress invasion and metastasis via inhibiting the metastatic protein expression such as MMPs (matrix metalloproteinases). The metastasis process is supported by neo-angiogenesis, and, interestingly, propolis has been found to hinder neovascularization [23,24,25]. Moreover, propolis revealed its ability to inhibit inflammatory mediators including tumor necrosis factor alpha (TNF-α), inducible nitric oxide synthase (iNOS), cyclooxygenase-1/2 (COX ½), lipoxygenase (LOX), prostaglandins (PGs), and interleukin 1- β (IL1-β) and other cancer inflammatory mechanisms [26]. As significant progress in elucidating the molecular mechanisms underlying the anticancer properties of propolis has been recently achieved, this review will focus on presenting the molecular targets and pathways involved in the anticancer effects of propolis.

## 2. Chemical Constituents, Bioavailability, and Biological Activities of Propolis

### 2.1. Chemical Constituents of Propolis

Propolis is a dark green, brown, or red resin derived from a varying mixture of flowers, wax, and bark exudates [27]. When cold, the substance is solid and brittle, but it turns soft and sticky at high temperatures. The chemical properties of its constituents vary extensively depending on the geographical region, plant species, climates, and season of collection [28,29,30]. Using different fractionation and separation techniques, studies have identified over 300 compounds in propolis collected from different regions [31]. This variety of chemically active molecules accounts for the various pharmacological properties and health-related uses of propolis extracts (Figure 1). Table 1 summarizes the most common chemical groups found in the propolis and examples of its active compounds. 

Collectively, the groups of compounds that consistently make up propolis are polyphenols such as flavonoids and phenolic esters, phenolic aldehydes, as well as ketones. Examples of flavonoids and flavonols commonly found in propolis include quecentin, islapinin, chrysin, alnusin, pinocembrin, naringenin, pinostribon, and ermanin [32,33]. Derivatives of benzoic acid such as gentisic acid, protocatechuic acid, salicylic acid phenylmethyl ester, and gallic acid also occur. Benzaldehyde compounds, including vanillin, protocatechualdehyde, and caproic aldehydes are also found in propolis [34,35]. Moreover, derivatives of cinnamic acid such as caffeic acid, isoferulic acid, and cinnamic acid methyl ester are also part of the mixture. Eicosine, tricosane, heneicosane, and other aliphatic hydrocarbons also constitute propolis in varying locations. In addition, sugars such as fructose, glucose, glucitol, and talose are constant components in the propolis, as are vitamins B1, B6, C, and E. Amino acids, including alanine, cysteine, butyric acid, isoleucine, leucine, valine, and others are commonly found in the resinous mixture. Similarly, esters as methyl palmitate, phenylethyl caffeate, benzyl benzoate, ethyl palmitate, tetradecyl caffeate, stearic acid methyl ester, and others are also present in varying combinations and quantities. Moreover, alcohols and ketones such as benzyl alcohol, coumarine, and xanthorrhoeol are consistent constituents of the extracts. Minerals such as sodium, zinc, and lead also occur in propolis. Other chemical groups that are often separated from the mixture are enzymes, waxy acids, fatty acids, and aliphatic acids [36].

The percentage of the different groups of compounds varies and are complemented by resins, bee wax, pollen, oils, and other organic substances. The most significant determinants of these disparities are the location (origin) and the local plants from which the constituents are derived. The poplar tree is the main source of propolis exudates, and it grows in New Zealand, many European locales, non-tropical parts of the Asian continent, and North America. Egyptian propolis also contains substances from this tree, along with caffeic acid esters and various alcohols. Russian, Brazilian, Cuban, Iranian, and Indian propolis all have slightly varying chemical constituents.

### 2.2. Bioavailability of Propolis and Its Constituents

An important aspect when studying the biological activities of various extracts derived from natural products is the evaluation of their bioavailability. The bioavailability of propolis is referring to the amount of propolis or its constituents able to reach systemic circulation and exert biological effects [49]. Curti et al. conducted a study to determine the bioavailability of a polyphenol mixture derived from brown propolis extract. They found that, after the oral administration of the brown propolis extract, rapid absorption and metabolization of the galangin into glucuronide occurred, and it was not accumulated in the mices’ livers. However, chrysin and its metabolites were not detectable after prolonged oral administration [50]. The low bioavailability of chrysin can be explained by its low water solubility and rapid metabolism [51]. In vivo studies were performed in rats to determine the pharmacokinetics and bioavailability of Algerian propolis extracts. They revealed that the bioavailability of the flavonoid naringenin was high following propolis administration compared to standard naringenin [52]. The bioavailability of ethanol extracts derived from Uruguayan propolis was also analyzed. They detected the free form of the main propolis flavonoids (pinobanksin 5-methyl ether, pinobanksin) in the urine of propolis-treated rats. These data presented strong evidence of the metabolism and circulation of the flavonoids in the body [53]. An in vitro study was conducted to identify bioavailable contents in Brazilian propolis. They studied the bioavailability of the propolis on the human colon adenocarcinoma cell line Caco-2 and the human hepatic carcinoma cell line HepG2. The results suggested that Artepillin C, which is a major part of Brazilian propolis, is a bioavailable antioxidant and could be used in the treatment of many diseases [54]. Despite the low bioavailability of Artepillin C, a compound with a wide variety of physiological activities, a unique dynamic binding interaction with human serum albumin was found to determine its mode of transportation in the body. This mode of transportation, in turn, determines Artepillin C’s distinct biological and physiological activities [55]. Additionally, the bioavailability of Artepillin C was tested in vivo. The results indicated that, following oral administration of Artepillin C, absorption and metabolization occurred and it was enough to produce biological effects [56]. Interestingly, a recent study by Boufadi et al. reported the variations in the bioavailability of crude propolis in comparison to pure phenolic or flavonoid compounds. It has been shown that the bioavailability of the pure compounds (phenolic acids, caffeic acid, kaempferol, and quercetin) was significantly less compared to the crude propolis [57]. Recently, the encapsulation of propolis extracts within polymeric nanoparticles, such as chitosan-Arabic gum and chitosan-folic acid, has been demonstrated to improve the bioavailability of the constituents of propolis. This approach shows promise in addressing the issue of low bioavailability that is often associated with some potentially active compounds found in propolis [58,59]. Overall, these results indicated that propolis and its constituents are bioavailable compounds, which makes them the ideal candidates for the development of new therapeutic agents.

### 2.3. Biological Activities of Propolis

Due to its multiple effects on human health and biological processes, propolis has historically been used for many purposes and recently several studies have shown that propolis demonstrates different biological activities. These activities, which are illustrated in Figure 2, include antibacterial [60,61,62,63,64,65,66,67], antiviral activities [68,69,70,71], antifungal [72,73,74,75], immunologic and inflammatory responses [76,77,78], and antioxidant activities [79,80,81,82,83,84,85].

## 3. Review Studies on Propolis Anticancer Activity 

Several studies have reported the anticancer effects of propolis and its active compounds. In in vitro studies, propolis has shown activity against a range of human cancer cell lines, including oral, gastric, cervical, colon, leukemia, stomach, skin, breast, and prostate cancers [14].

Most of these studies have examined the effects of propolis extracts from various geographical origins on the growth, proliferation, and metastasis of cancer cells in vitro. In this section, we reviewed the anticancer effects of propolis extracts from different geographical origins. 

Salem et al. investigated the anticancer activity of Egyptian propolis in vitro, as well as its protective role against methotrexate (MTX) toxicity, administered in an Ehrlich ascites carcinoma (EAC) experimental model. The results of the study showed that ethanolic extract of propolis (EEP) was cytotoxic to cancer cell lines and showed antitumor effects against EAC model by reducing the volume of tumors, decreasing the count of viable cells in the mass, and prolonging the life span of affected mice. In addition, improvements in methotrexate-induced hepatorenal toxicity in animal models were also recorded. Egyptian propolis extract revealed high apoptotic effects through an increase in BAX (pro-apoptotic protein), caspase-3, and cytochrome-c expression levels, and by a reduction in B-cell lymphoma2 (BCL2) (antiapoptotic protein) in the tested cancer cell line [16]. Propolis also enhanced the G0/G1 cell cycle arrest induced by methotrexate. Similar findings were reported by El-khawaga et al. in in vitro analysis of Egyptian propolis [19].

Using both in vitro and in vivo models to assess the tumor suppression characteristics of Philippine propolis, a study was carried out by Desamero et al. In their study, in vitro exposure of propolis to human gastric cancer (GC) cell lines, namely, AGS, NUGC-4, MKN-45, and MKN-74, showed that differentiated-type GC cell lines (AGS, NUGC-4, MKN-45) were more sensitive to the EEP revealed by a decrease in cancer cell proliferation in a time-and-dose dependent manner compared to the MKN-74 that appeared to be resistant to the EEP exposure. The researchers of this study also investigated further the possible mechanisms by which EEP perceived its anticancer activity. They revealed that EEP acts in vitro through modulation of cell cycle and apoptotic-related genes, resulting in G0/G1 phase arrest. However, this effect was only observed in the differentiated tumor but not in the anaplastic type. The in vivo component of the research involved the use of animal models: a differentiated type of gastric adenocarcinoma *A4gnt* KO mice as well as C57BL/6J mice (matched wild type). The two groups were exposed to EEP or distilled water intragastrically for 30 days. The researchers showed a significant propolis-induced antitumor effect revealed by gross and histologic mass regression in the EEP-treated group and found that EEP exposure in the animal models affected cell cycle machinery [86].

Turan et al. measured polyphenolic and flavonoid contents, antioxidant properties, and cytotoxicity of Turkish EEP. Using spectrophotometric analysis, the total polyphenolic and flavonoid concentrations in the extract were 124.6 mg gallic acid per gram and 42 mg quercetin/g of dry weight sample, respectively. Furthermore, the investigators found that propolis has potent cytotoxic effects on human cancer cell lines, liver, colon, breast, prostate, and cervical cancers, with the highest cytotoxicity being observed in prostate cancer cells [15].

The antiproliferative potential of Algerian propolis extracts was investigated by Brihoum et al. They injected albino Wistar rats with 25 mg/kg of the propolis extract and evaluated its effect on a benzopyrene-induced lung carcinogenesis model. They observed a dose-dependent blockage of respiratory cell adhesion usually driven by fibrinogen. Concurrently, propolis inhibited oxidative stress generated by the carcinogenic agent [87]. Histologic analysis of lung tissues confirmed the chemoprotective effects of EEP. Similarly, Doi et al. found significant propolis-driven suppression of tumor size and aggressiveness of colorectal cancers induced by 1,2-dimethylhydrazine (DMH) injection [26]. This reduction in inflammation-associated tumorigenesis was linked to the anti-inflammatory effects of EEP, which were indicated by significant reduction of inflammation-related molecules such as iNOS, TNF-α, and glutathione peroxidase-2.

An in vitro study conducted by Choudhari et al. examined the antitumor effects EEP from Indian stingless bees on cancer cells derived from human breast cancer (MCF-7), murine melanomas (B16F1), human colon adenocarcinoma (HT-29), and human epithelial colorectal adenocarcinoma (Caco-2). The extract’s cytotoxicity was measured by MTT and Trypan blue dye exclusion assays. The authors found that the IC50 at all tested cell lines was 250 μg/mL of EEP [88]. Moreover, microscopic features of apoptosis were observed in all cells following 24 h of exposure to propolis. 

Calhelha et al. compared the activity of Portuguese propolis extracts against human cancer cell lines with its cytotoxicity to normal cell lines. They revealed that aside from colon carcinoma, the specificity of EEP to cancer cells is indistinguishable from that towards normal cells at 50% inhibitory concentrations [89]. The authors recommended sufficient in vivo experimentation to identify constituents with specific antitumor effects and to determine optimal therapeutic doses. In contrast, a comparison of the activity of Thailand propolis on normal and cancerous cells carried out by Umthong et al. found significant differences with the propolis showing cytotoxicity against cancerous but not normal cells. In the experiment, five cancer cell lines (KATO-III, BT474, Chago, Hep-G2, and SW620) and two normal cell lines (liver and fibroblast) were exposed to different concentrations of EEP. The researchers found that the propolis had no effect on the viability of the fibroblasts and hepatocytes, yet it exhibited significant toxicity against all the five malignant cell lines [90]. Using MTT assays, Amini-Sarteshnizi et al. investigated the effect of extracts from Iranian propolis on human gastric adenocarcinoma cell line (AGS). Consistent to several other studies, they found a dose- and time-dependent antiproliferative effect on the cells by the EEP treatment [91].

In an experimental determination of the phytochemical profile, antioxidant properties, and antitumor effects of Brazilian red propolis, de Mendonça et al. fractionated EEP and measured its effect on colon, ovary, and glioblastoma cell lines. They revealed that, in addition to potent antioxidant capacity, propolis exhibited significant cytotoxicity against the three cancer cell lines [92].

Additionally, Mostafa et al. investigated the in vitro and in vivo anticancer activity of Omani propolis against colorectal tumors [93]. The experimental setup consisted of a rat model assessing the extract’s ability to suppress azoxymethane-induced colon cancer and an in vitro determination of EEP’s cytotoxicity of colorectal carcinoma cell lines [93]. In both settings, significant anticancer effects of the extract were recorded.

Using oil from edible vegetables, Carvalho et al. extracted active compounds from propolis and assessed their composition using chromatographic methods. Then, they tested their activity against cell lines of leukemia (HL-60), colon (HCT-8), breast (MDA-MB-231), and SF-295 (brain) cancers [94]. The mixture of phenolics and flavonoids identified in the extract demonstrated noteworthy activity against all tumors in both in vivo and in vitro settings. Collectively, the components were more bioactive than they were individually.

Li et al. examined the relationship between structural composition and antitumor activity in Myanmar propolis [95,96]. They isolated 13 tritepenes and 4 flavanones from the propolis and exposed them to 26-L5 colon, B16-BL6 melanoma, A549 lung, and HeLa cervical cancer cell lines. All components showed significant antitumor activities compared to controls, albeit with different potencies [95,96,97,98]. The authors proceeded to suggest structure-activity relationships that may account for the anticancer effects. In a similar study by Li and others, a methanolic extract from Myanmar propolis was investigated for activity against PANC-1 pancreatic cancer cells lines. In addition to identifying an additional highly bioactive tritepene from the methanolic extract, the authors found that the methanolic extract of propolis exhibited significant cytotoxicity against the tested cell lines [95]. Moreover, they observed morphological changes consistent with apoptosis in the cells after 24 h of exposure.

Utispan et al. evaluated the cytotoxic effects of a previously unstudied stingless bee species, *Trigona Sirindhornae*, on HN30 and HN31 cell lines from head and neck growths. The investigators treated separate samples from primary and metastatic masses with propolis extracts and recorded their viability after exposure to different concentrations of the extracts [99]. They observed significant decreases in both primary and metastatic neoplasms from the head and neck region.

Barlak et al. used spectrophotometry and proteomics analysis to assess the impact of an aqueous extract of Turkish propolis on PC-3 prostate cancer cell lines. The results revealed that propolis extract significantly altered the protein expression profile of the prostate cancer cells [100]. They attributed the anticancer impact of propolis to its antioxidant properties.

Khacha-Ananda et al. determined the phenolic and flavonoid contents and the cytotoxic and antioxidant capacity of Thailand propolis. The extraction methods used to isolate components of the extract were maceration and sonication, with the latter showing less efficiency but yielding extracts with higher antioxidant potential. Moreover, the extract demonstrated noteworthy cytotoxic effect on HeLa and A549 cancer cell lines [101]. The experiment’s findings are relevant to considerations of the most appropriate extraction technique for propolis constituents for anticancer and antioxidant purposes. 

### Propolis Constituents Exhibiting Anticancer Activity

The anticancer activities have been attributed to several active compounds found in extracts of propolis from different regions, with the most frequently identified being caffeine acid phenethyl ester (CAPE) and chrysin. CAPE is the most studied compound of propolis in in vitro studies and is thought to be responsible for its several biological activities. Elucidation of its role in the anticancer activity of propolis was reported by Orsolic et al., who demonstrated an increase in apoptosis rates up to 31.24% in fibrosarcoma cell lines following exposure to CAPE [102]. Moreover, Lee et al. observed that analogues of CAPE exhibited marked antitumor activity against fibroblasts derived from oral submucous fibrosis and oral squamous cell carcinoma [103]. Additionally, in vitro studies conducted by Chen et al. (effect of CAPE on leukemic cell lines) [104], Nomura et al. (inhibition of cell transformation and induction of cell death by CAPE) [105], Lee et al. (role of p53 and p38 MAP-kinases in cell death induced by CAPE) [103], Hung et al. (cell death caused by CAPE and its amide and ester derivatives on cervical cancer cell lines) [106], Jin et al. (mitochondria-mediated cell death induced by CAPE in leukemic cell lines) [107], and Watabe et al. (NF-κB inhibition and Fas activation by CAPE induce apoptosis in breast cancer cells) all demonstrated a cytotoxic role of CAPE in several cancer cell lines [108]. An elaborate modulation of mitogen-activated protein kinases (MAPK) by CAPE as a mechanism of apoptosis induction has also been identified by Bulavin et al., Sanchez-Prieto et al., and Weng et al. [109,110,111]. Ishida et al. found that due to instability of the active constituent, CAPE, the tumor suppression of propolis quickly diminished during in vivo experimentation [112]. However, the activity was regained following a combination of CAPE with γ-cyclodextrin (γCD), which increased its stability in an acidic setting.

Bhargava et al. compared the cytotoxic activity of CAPE, its equivalent in Brazilian propolis extracts known as artepillin C (ARC), and green propolis-supercritical extract (GPSE). Like CAPE from New Zealand propolis, ARC caused cell death by destroying mortalin-p53 complexes, resulting in p53 activation and subsequent growth cessation. Additionally, cell viability assays revealed that GPSE had higher cytotoxicity than ARC, an observation that was also made in the comparison between γ cyclodextrin-complexed GPSE and ARC [113]. The authors concluded that the GPSE-γCD combination may be an effective anticancer therapeutic strategy. Kuo et al. recorded significant inhibition of C6 glioma cells growth in vitro and in vivo from CAPE extracts exposure [114].

Wadhwa et al. studied the molecular mechanisms mediating the cytotoxic and antimetastatic characteristic of CAPE, the most active compound in New Zealand propolis. Using microarrays and molecular docking analysis, the researchers found that CAPE upregulates GADD45α and p53, both of which are key regulators in tumor suppression pathways. Moreover, the study revealed that CAPE disrupted complexes between mortalin and p53, which induced its nuclear translocation and restored the activity of p53 and subsequent cell cycle arrest [115]. The investigators also identified a downregulation of mortalin and several other regulators that play a role in tumor metastasis. Additionally, they provided experimental evidence for using a combination of cyclodextrin (γCD) with CAPE to slow down its degradation into caffeic acid, which is caused by secreted esterases in in vivo models.

Chen et al. investigated the molecular basis of CAPE-induced cell death in human leukemic cells. The study showed that following rapid intracellular entry of CAPE, the compound induced a concentration- and time-dependent inhibition of cell survival [104,116,117]. Characteristic fragmentation of DNA along with unique morphological changes that marked apoptosis were also observed. The mechanisms underlying these changes included activation of caspase 3, reduction of BCL2 expression, and BAX upregulation. A similar decrease in BCL2 expression by EEP was observed in an in vitro experiment conducted by Sadeghi-Aliabadi, Hamzeh, and Mirian, who compared its activity against prostate and bladder cancer cells [118]. However, this study did not record significant increases in the levels of p53 in cells treated with propolis extracts. Other in vitro studies that showed a downregulation of BCL2 and simultaneous activation of pro-apoptotic molecules such as BAX include those by Lee et al. (role of p53 and p38 MAPK in glioblastoma cell death caused by CAPE) [103], Jin et al. (mitochondria-mediated cell death in myeloid leukemia cell lines treated with propolis) [107], and Watabe et al., who studied the inhibition of NF-κB and activation of Fas as mechanisms of apoptosis in CAPE-treated breast cancer cells [108]. 

Chrysin is another major component present in propolis that has been identified as a prominent player in various pathways involved in the antiproliferative activity of the propolis. Woo and co-authors demonstrated that chrysin caused cell death in leukemic cell lines (U937), downregulating the PI3K/Akt pathway [119]. Moreover, the compound inactivates NF-κB and IAP, which is an effect that results in caspase 3 activation and subsequent initiation of apoptosis. In another study, Jafari-Ghahfarokhi et al. assessed the effect of chrysin, CAPE, and ethanolic extracts of propolis on the expression of phospholipase D1 (PLD1) gene in cultured gastric cancer cell lines [120]. They found that the compounds inhibited cell growth by downregulating the PLD1 gene expression. The inhibitory effect was concentration dependent for all components of EEP. Additionally, Li et al. concluded that chrysin stimulates the release of TNF-α by suppressing NF-κB, causing cell death of tumor cells [98]. Moreover, Touzani et al. studied the potential activity of pinocembrin, quercetin, and chrysin components of Moroccan propolis as multitarget therapeutic agents [121]. The authors found that these compounds mediated significant cytostatic, immunomodulatory, and antiproliferative effects. Using flow cytometry to monitor changes in viability of cell populations, Samarghandian et al. assessed the effect of chrysin on PC-3 prostate cancer cells. The study revealed a time-sensitive antiproliferative effect of the chrysin on cancer cells that peaked after 72 h [122].

## 4. Molecular Mechanisms of Anticancer Activities of Propolis

### 4.1. Apoptosis Induction

The evidence for apoptosis as the principal cellular mechanism through which propolis exerts its antitumor effects has been derived from several in vitro studies using several human cancer cell lines. Propolis could trigger both intrinsic and extrinsic apoptosis signaling pathways. Active molecules in the propolis extract stimulate apoptosis via modulating regulatory proteins (pro- and anti-apoptotic proteins). For instance, the apoptotic effect of propolis was found to be mediated through activation of pro-apoptotic proteins such as Puma and Bax in human oral cancer cell lines [17]. In a study aimed at characterizing polyphenols in propolis extracts and their anticancer effects, Czyżewska et al. used a combination of chromatography and mass spectrophotometry to separate EEP into compounds such as chrysin, pinocembrin, caffeic acid, galangin, and ferulic acid. Then, they examined the individual and collective effects of these components on the viability of human squamous cell carcinoma cell lines of the tongue (CAL-27). MTT assays indicated that the extracts displayed a dose-dependent cytotoxicity against the tested cell lines, which was more effective when used in combination [18]. The molecular mechanism of this effect was shown to be through activation of caspases 3, 8, and 9, key regulatory proteins in apoptotic pathways, modulation of kinase C, and suppression of tyrosine kinase signaling pathways [18]. In another study also using CAL-27 cells, propolis and its components were shown to trigger PRODH/POX-dependent apoptosis [123]. Ethanolic extract of propolis was also shown to cause apoptotic cell death in colon cancer cell line by inducing DNA damage [124]. Studies have also shown that propolis induced apoptosis in the SW620 human colorectal cancer cell line through mitochondrial dysfunction caused by high production of reactive oxygen species (ROS) and caspase activation [125,126]. Galangin- and chrysin-induced apoptosis and mitochondrial membrane potential loss in B16-F1 and A375 melanoma cell lines were mediated through p38 mitogen-activated protein kinase (MAPK) and Bax activation, as well as downregulating the ERK1/2 signaling pathway [127,128]. Brazilian propolis and its component CAPE triggered apoptosis in prostate cancer by downregulating apoptosis inhibitor proteins 1/2 (cIAP-1/2) [97,129]. Several other studies used breast cancer cell lines and prostate cancer cell lines models and demonstrated that propolis could trigger apoptosis through the same mechanism across different cancer cell types, which has been found to be through downregulating PI3K/Akt, p38 MAPK, and ERK1/2 signaling pathways, ER stress, and ROS production and subsequent loss of mitochondrial potential [130,131,132]. Further evidence of the apoptotic effect of propolis on cancer cells was established in a study by Orsolić et al., in which mice were injected with mammary carcinoma cells and administered caffeic acid, CAPE, and quercetin. The compounds were found to significantly decrease the number of viable foci of the tumor cells in rats [102]. The authors attributed this effect with propolis’ immunomodulatory properties as well as its induction of both apoptosis and necrosis.

Further investigations of the molecular mechanisms of the propolis’s apoptotic effect have revealed several possible participating molecules. One of these mechanisms is the induction of tumor necrosis factor-related apoptosis inducing ligand (TRAIL), which is a crucial selective mediator of apoptotic cell death through the death domain of its receptor [133,134,135,136]. Some cancers develop resistance to TRAIL-mediated cell death, a process that, as revealed in several in vitro studies, can be reversed by propolis-derived compounds. Treatment of prostate and colon cancer cells with propolis and its constituents was proven to cause apoptosis via a TRAIL-dependent mechanism [133,137]. Szliszka et al. found that propolis extracts at concentrations of 50 μg/mL significantly increased the levels of TRAIL in cervical tumor cell lines. Szliszka et al. investigated the impact of Brazilian propolis treatment on TRAIL-resistant LNCaP prostate cancer cells [136]. Using a combination of flow cytometry, fluorescent staining, and HPLC techniques, they recorded an increase in the expression of the TRAIL-R1 and TRAIL-R2 death receptors on the cells, which translated to a significant increase in apoptosis rates in the treated cell population. In a different study, Szliszka et al. evaluated the role of TRAIL in propolis-induced death in cancer cells [135]. The researchers obtained TRAIL-resistant HeLa cell lines and exposed them to phenolic acids extracted from propolis [135]. The population of cancer cells displayed an increase in the rate of apoptosis up to 71%, with the most active concentrates driving this effect being CAPE and apigenin. A similar conclusion was reached by Szliszka et al., who examined the effect of the propolis on the viability of LNCaP prostate cells. Using MTT assays and flow cytometry, the investigators monitored both viability and apoptotic rates in the cancer cells [135]. They also assessed the expression of the TRAIL death receptor during the exposure period. They found that propolis sensitized cancer cells to apoptosis by upregulating TRAIL-R2. Moreover, experimentation with HeLa cell line by Szliszka et al. indicated that propolis caused an increase in TRAIL-mediated cell death [135].

Suk et al. described a role for activating factor 3 (ATF-3) in inducing cell death in Hep3B and HepG2 hepatoma cells exposed to Taiwanese propolis. The researchers found that the propolis not only caused concentration-dependent apoptosis but also an increase in the stress-inducible ATF-3 and caspases 3 and 9 [138]. Similarly, McEleny et al. described a role of caspases in apoptotic death of PC-3 cells exposed to the propolis extract [129]. 

Xuan et al. investigated several molecular mechanisms of propolis-induced cell death in MCF-7 and MDA-MB-231 breast cancer cells. After treating these cell lines with Chinese propolis, the researchers monitored the levels of annexin v, p53, NF-κB, and ROS. These molecules were found to be elevated following exposure of the cells to the alcoholic extract of the propolis [139]. Moreover, propolis caused a rise in mitochondrial membrane potential, further supporting the involvement of mitochondria in the apoptotic effect of propolis. In a 2011 study, Xuan et al. examined the molecular events behind propolis-mediated cell death in umbilical vein endothelial cells. The investigators revealed that at high concentrations, propolis increased the amounts of integrin β4, ROS, and p53 [140]. The high expression levels of these molecules, in turn, drove a decrease in mitochondrial membrane potential, which is an important step in apoptosis.

Aso et al. treated U937 leukemia cells with the propolis extract and studied its impact on the DNA. They found that propolis not only repressed DNA replication but also inhibited synthesis of RNA and proteins [141]. The effect on DNA synthesis was partially irreversible and its fragmentation was prevented by a caspase inhibitor. Further indication of the role of both caspases in the cytotoxicity of propolis was demonstrated by Chen et al. [117]. In their 2008 study, pancreatic cancer cell lines BxPC-3 and PANC-1 were incubated with propolis extracts and the cell viability monitored over time using trypan blue dye exclusion. In addition to quantitative inhibition of viable cells, the propolis extract induced DNA fragmentation and G1 cell cycle arrest [117]. The association of this effect with caspase was underlined by a doubling in the activity of caspases 3 and 7. In addition, the cytotoxicity, along with the morphological changes, were diminished by a pan caspase inhibitor. Similarly, another 2004 study by Chen and co-authors assessed the radical scavenging and apoptotic activity of Taiwanese propolis. The researchers treated A2058 melanoma cells with the propolis extract, after which they evaluated cells viability, morphologic changes, and caspase activity [116]. They recorded significant suppression of proliferation, cell cycle arrest at G1 phase, and an increase in caspase 3. Furthermore, the activity of death receptors such as Fas and Fas-L was shown to be activated to induce apoptosis in tumor cells. Among the most prominent apoptotic mechanisms, however, is the reactivation of tumor suppressor genes, particularly p53, which is a regulator of cell cycle and apoptosis. Figure 3 represents a model summarizing the molecular mechanisms of how propolis induces apoptosis in cancer cells.

### 4.2. Autophagy Induction

Autophagy is a mechanism by which cells degrade their constituents such as proteins and small organelles in a protective manner to keep homeostasis. Such a mechanism helps in protecting normal cells from different diseases. In cancer cells, autophagy can act as both a cell survival and a cell death mechanism, depending on the pathway [142,143]. Here, we describe the effect of propolis in modulating the autophagy process in cancer cells. A recent study showed that Chinese propolis and its major constituent, CAPE, were found to induce autophagy in a breast cancer cell line (MDA-MB-231) through upregulating LC3-II and downregulating p62, which are two major markers in autophagy pathway [144]. Another study also showed autophagy induction by Chinese propolis in the melanoma A375 cell line. By examining autophagy markers, the conversion of LC3-I/LC3-II, Atg5/Atg12 complex, and p62 all increased after treatment with Chinese propolis. Additionally, beclin-1 was downregulated in these cells, suggesting the induction of autophagy [145]. Therefore, the anticancer effect of propolis via autophagy induction can serve as a cell death mechanism to stop cancer cell proliferation. Conversely, a recent study showed that the induction of autophagy by Brazilian propolis constituent artepillin C (ArtC) promoted cancer cell survival and reduced its sensitivity to anticancer drugs. It was shown that in CWR22Rv1 prostate cancer cell lines that induction of autophagy served as a protective mechanism for cancer cells, but when these cells were treated with autophagy inhibitors, apoptosis was augmented and more cell death was recorded, suggesting that combining ArtC and autophagy inhibitors can be used to treat prostate cancer [146]. 

In summary, the role of autophagy in cancer treatment is yet to be defined and more studies are required to understand the molecular mechanisms through which propolis affects autophagy and how this can be related to cancer treatment. 

### 4.3. Anti-Proliferative and Cell Cycle Arrest

Cell cycle progression involves two consecutive events, mainly characterized by genomic DNA replication and the segregation of replicated chromosomes into daughter cells [21]. Defects developing in cell cycle checkpoints is one of the main mechanisms underlying tumorigenesis [147]. Evidence for the role of cell cycle control in the anticancer effect of propolis was reported by several studies. Li et al., investigated the effect of a Brazilian propolis on DU145 and RC58T/h/SA#4 cell lines of prostate cancer. Besides reporting significant growth inhibition, the investigators traced the activity to a combination of S phase arrest, G2 regulation, and concurrent downregulation of cyclins D1, cyclin-dependent kinase 4 (CDK4), and cyclin B1, major regulators in the cell cycle [97]. Gunduz et al. described similar cell cycle arrests and telomerase shortening induced by propolis on U93 leukemia cells [148]. Moreover, in an in vitro experiment involving SW480 cells from colorectal cancer, He et al. identified comparable CAPE-induced cell cycle arrests and downregulation of β-catenin signaling, which is an effect that was also reported by Xiang et al. [149,150]. In oral cancer, propolis and its active component CAPE inhibited the growth of oral cancer cell lines through modulating cell cycle regulator proteins cyclin D, Cdks-2/4/6, and cyclin-dependent kinase inhibitors, thus causing cell cycle arrest at G2/M phase [151]. Likewise, genistein, a common component found in propolis, was reported to induce cell cycle arrest at G2/M in human colon cancer cells through upregulating a number of tumor suppressor genes such as p53 [152]. Using the same human colon cancer cell line, Shimizu found CAPE-induced G0/G1 cycle arrest attributable to Cip1/p21 stimulation by artepillin C in HCT-8 colon cancer cells [153]. Similarly, propolis caused cell cycle arrest (G0/G1 and G2/M phase) on BT-474 breast cancer cells via upregulating p21 and p27 expression [154]. Ethanolic extracts of Brazilian propolis treatment of prostate cancer cells significantly arrested these cells by altering the expression of cyclin A, B, and D1, and Cdk as well as p21 [97]. Moreover, it has been reported that different types of propolis and its components exhibited anti-proliferative effect via triggering cell cycle arrest by enhancing the expression of various tumor suppressor genes including p53, p16, and Rb as well as downregulation of oncogenes (MIFT and K-Ras) [152,155]. Figure 4 represents a model summarizing the molecular mechanisms of how propolis induces cell cycle arrest in cancer cells.

### 4.4. Anti-Metastatic and Anti-Angiogenesis Effects

Various studies have shown that propolis and its components demonstrated effective anti-metastatic and anti-angiogenic properties. These zinc-dependent proteolytic enzymes play a pivotal role in degrading the extracellular matrix to enable tumor cells to spread to a new site. Hwang et al. investigated the impact of CAPE on metalloproteinases gene expression in HT1080 fibrosarcoma cells [156]. They found that propolis inhibited metalloproteinase activity, reduced cell motility, migration, as well as colony formation by cancer cells. Propolis was shown to inhibit cancer cell metastasis and angiogenesis by decreasing the activity of MMPs-2/9 [23]. The reduction in MMPs activity was traced to the downregulation of growth factors like EGF, vascular endothelial growth factor (VEGF) via modulating Jun N-terminal kinase, ERK1/2, NF-κB, and Akt signaling [23,24]. A recent study by Frión-Herrera et al. (2020) showed that Cuban propolis and its active component nemorosone inhibited cell migration via altering the expression of E-cadherin, vimentin, and β-catenin in HT-29 and LoVo colorectal cell lines. Similar anti-angiogenic and anti-metastatic effect of propolis were revealed in the breast cancer cell line model [157]. The contribution of angiogenesis inhibition in propolis-driven anticancer activity was also explored by Ahn et al. using ICR mic models where they assessed the impact of Brazilian propolis on the formation of new blood vessels. They revealed that the Brazilian propolis extract has significant antiangiogenic effects, which was largely attributed to artepillin C [158]. Considering the role of angiogenesis in the implantation of metastatic cancers and in the vascularization of neoplastic masses, the inhibition of this process by propolis represents a promising anticancer mechanism. Daleprane et al. explained the antiangiogenic properties of propolis as being mediated by a reduction in the levels of hypoxia-inducible factor-1α (HIF-1α). The authors tested this effect in in vivo samples of embryonic stem cells. The results showed that propolis caused a decrease in HIF-1α, which in turn induces a reduction in the expression of VEGF gene [159]. Similar results were obtained by Meneghelli et al., who examined the effect of Southern Brazilian autumnal propolis on tubulogenetic potential of endothelial cells. The study revealed that propolis extracts decreased proliferation of the endothelial cells and, consequently, reduced new vessel formation [160]. Altogether, these findings suggest that an important mechanism through which propolis exerts antimetastatic effects is through inhibition of metalloproteins, downregulation of EGF, HIF-1α, and VEGF via modulating several signaling pathways including, c-Jun N-terminal kinase, ERK1/2, NF-κB, and Akt signaling. Figure 5 represents a model summarizing the molecular mechanisms of how propolis suppresses metastasis in cancer cells.

### 4.5. Suppressing Inflammatory Pathway

The anti-inflammatory effect of propolis accounts for part of its anticancer properties, especially as far as inflammation-associated tumors are concerned. For instance, a study by Chiu and his team showed that propolis enriched by CAPE inhibited cyclooxygenase-2 subsequently reduced prostaglandin 2 synthesis in human oral epidermal carcinoma KB cells [25]. Similarly, Ozturk and others have reported that propolis and its component CAPE exert anticancer activity through modulating the inflammatory response. This response is associated with suppression of COX-2/LOX, pro-inflammatory cytokines, and NF-κB signaling with an increase in the production of anti-inflammatory cytokines (IL-10) in the melanoma cell line [161]. Moreover, propolis was reported to alter the immunological response through activation of macrophages, natural killer, and T-lymphocytes. Using gastric cancer and breast cancer models, propolis targeted the inflammatory cascade by downregulating Toll-like receptor 4 (TLR-4), glycogen synthase kinase 3 beta (GSK3 β), and NF-κB signaling pathways [144,157,162]. In a 2018 study, Zheng et al. demonstrated that Chinese propolis exerts anticancer effect in human melanoma cells via trigger apoptosis, cell cycle arrest, and autophagy by targeting the NLRP1 inflammatory pathway [145]. 

In an in vivo setting involving mice with Ehrlich ascites tumors, Orsolić and Basic assessed the effect of water-soluble constituents of Croatian and Brazilian propolis on the volume of peritoneal exudates released by the cancer and the macrophage activity against it. In the exposed animals, the researchers not only recorded decreased ascites exudation but also enhanced macrophage activity, suggesting a propolis-driven synergistic action and enhancement of the potency of the innate immunity cells [163]. A similar intersection involving NF-κB suppression has also been found to underlie both immunomodulatory and anticancer properties of the propolis. In a study on the anti-inflammatory role of CAPE, Wang et al. assessed its effect on the activity of monocyte-derived dendritic cells. They found that, along with several other mediators of inflammation, NF-κB activation was decreased by CAPE treatment [164]. This mechanism has also been identified by several other studies exploring the antitumor activity of propolis, representing an important overlap between the two areas of activity.

### 4.6. Epigenetic Modulations 

MicroRNAs (miRNAs) play an important role in cell proliferation, apoptosis, development, and differentiation. Recently, Misir et al. have demonstrated that Turkish propolis markedly alters the miRNA expression of tumor suppressors’ genes (miR 34, 15a, and 16-5p) as well as the miR 21 and breast cancer gene (BRCA ^1/2^) in human breast cancer MCF-7 cells [165]. Interestingly, Omene et al. demonstrated that CAPE causes epigenetic modification of genes involved in cell growth, chemoresistance, as well as angiogenesis. This effect arises from the compound’s inhibition of histone deacetylase, causing an increase of acetylated proteins in breast cancer cells. These changes induce alteration in the expression levels of estrogen, progesterone, HER2, and EGFR receptors in the MDA-MB-231 breast cancer cell lines [166]. The effect of another Turkish propolis called “Aydin” was evaluated on glioblastoma (GBM) and brain cancer stem cells (BCSCs). The propolis treatment regulated large numbers of miRNAs in both cell types. In this study, they identified the role of two new miRNAs, miR-30d-5p and miR-335-5p, that were not reported previously in brain cancer studies. The propolis treatment decreased the expression of miR-30d-5p, a potential oncomir in GBM and increased the expression of miR-335-5p, a potential tumor suppressor in GMB [167]. Altogether, these findings suggest that propolis may have a reverse effect on cancer cell proliferation and progression by targeting oncogenes and tumor suppressor genes or altering miRNA expression.

### 4.7. Telomerase Inhibition

Telomerase enzyme is known to be upregulated in cancer cells which might contribute to cell survival by escaping cellular aging and maintaining the proliferation abilities. Therefore, a key target to cancer treatment is through inhibiting telomerase activity. Propolis was shown to inhibit the telomerase reverse transcriptase activity in leukemia cells. Even though the exact mechanism of telomerase inhibition is yet to be defined, it is thought that inhibiting telomerase activity will lead to a progressive telomere shortening which will eventually induce senescence and cell death [148,168]. More studies are needed to elucidate the mechanism of telomerase inhibition by propolis. 

Table 2 summarizes the molecular mechanisms by which propolis from different origins exerts its anticancer effect using different human cancer cell lines’ models.

## 5. Synergistic Effect of Propolis with Other Anticancer Agents

Propolis has been shown to increase the activity of existing chemotherapeutic agents and inhibit some of their side effects. Using glioblastoma cell lines and H3-thymidine incorporation to measure cell division, Markiewicz-Żukowska et al. assessed the impact of propolis on the activity of temozolomide against these cells. The results indicated that the combination of the two substances enhanced tumor suppression [187]. The mechanism of EEP’s growth inhibition seemed to be mediated by reducing NF-κB activity. In another study, Milosevic-Djordjevic et al. studied the in vitro activity of propolis against carcinogenesis in lymphocytes and breast cancer cells as well as how EEP in combination with mitomycin C affected the chemotherapeutic properties of the latter [188]. The findings revealed a concentration- and time-dependent anticancer effect of EEP, which was higher in cells exposed to a combination of propolis and mitomycin C. Padmavathi et al. also reported enhancement of paclitaxel cytotoxicity against breast cancer in female mice exposed to propolis [189].

Salim et al. compared tumor suppressive, antioxidant, and antimicrobial activity of Egyptian propolis and doxorubicin (a standard chemotherapeutic agent) [190]. The research results indicated that EEP, whether alone or in combination with doxorubicin, has higher antioxidant, apoptotic, as well as antiproliferative effects on prostate cancer cell lines compared to doxorubicin alone. In a similar study, Hasan et al. compared the effectiveness of a combination of EEP and extracts of *Curuma zanthorrhiza* in inhibiting proliferation of MDA-MB-231 breast cancer cell lines in an in vitro setting [191]. Using MTT assays, they found that the two extracts exerted a synergistic effect and exhibited higher antitumor activity than either of the extracts used individually. The combined treatment also had higher suppression activity than the doxorubicin control [191].

## 6. Conclusions

Along with other honey products, propolis has found a myriad of applications across cultures over time. In the scientific community, propolis has drawn interest following its immunomodulatory, antimicrobial, wound healing, hepatoprotective, and anticancer properties. A resinous mixture of over 300 chemical compounds, propolis has been the subject of many scholarly studies seeking to verify and understand its effects on various biological activities. A great deal of this interest is driven by the need to develop effective chemotherapeutic and chemopreventive agents; the urgency of which is underlined by the high morbidity and mortality rates accompanying many cancers. As a result of these needs, many in vitro and in vivo studies have been performed and many reports published that have documented both the anticancer effects and the underlying molecular pathways and signaling mechanisms utilized by propolis in various cell lines and animal models. These investigations have revealed such effects via inhibition of proliferation, decrease of metastatic potential, and direct cytotoxicity against cancerous cells. These effects are mediated through a collection of interdependent subcellular pathways and molecules, with the most apparent being apoptosis. Studies have identified roles of cell cycle arrests at different stages (mostly G0 and G1), modulation of mitogen-activated protein kinases (MAPK), and induction of TRAIL death receptor expression, mitochondrial membrane potential disruption, and regulation of caspases. Despite the large number of studies on the topic, a comprehensive understanding of the extent of effects of propolis on cancer cells, including the definite number and types of susceptible cancers, has yet to be established. Furthermore, this knowledge has hardly been extrapolated beyond in vitro and in vivo studies. Therefore, the knowledge base on propolis and its chemoprevention and treatment role, despite being an area of significant excitement and potential, has yet to be well established. Moreover, as research into the molecular basis of propolis effects on tumors continues, the role of more cellular pathways is becoming apparent.

## Figures and Tables

**Figure 1 pharmaceuticals-16-00450-f001:**
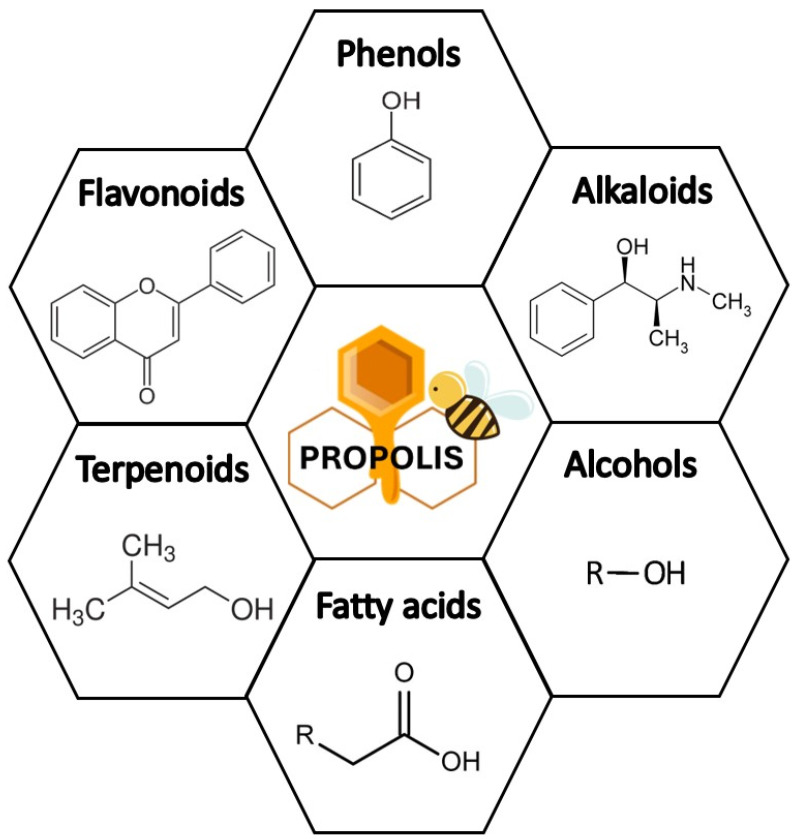
The main chemical groups responsible for the various pharmacological properties and health-related uses of propolis.

**Figure 2 pharmaceuticals-16-00450-f002:**
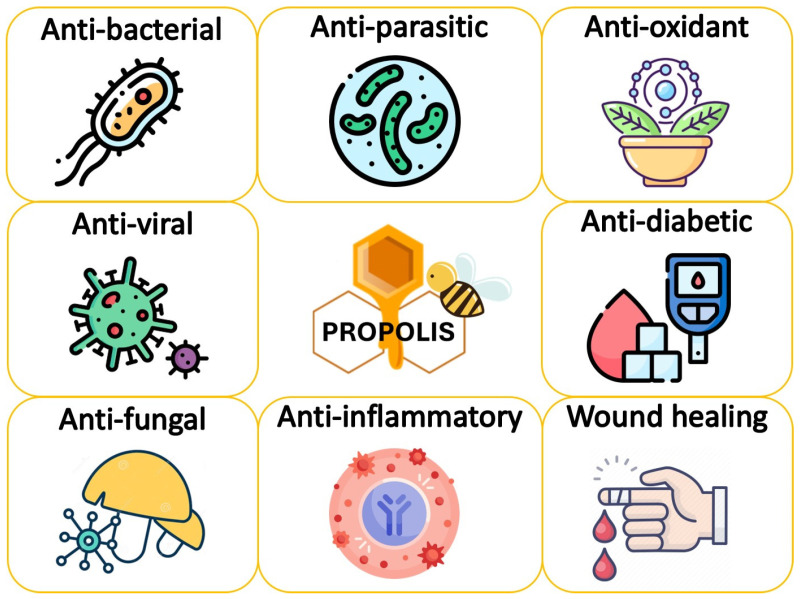
Summary of the main biological activities of propolis.

**Figure 3 pharmaceuticals-16-00450-f003:**
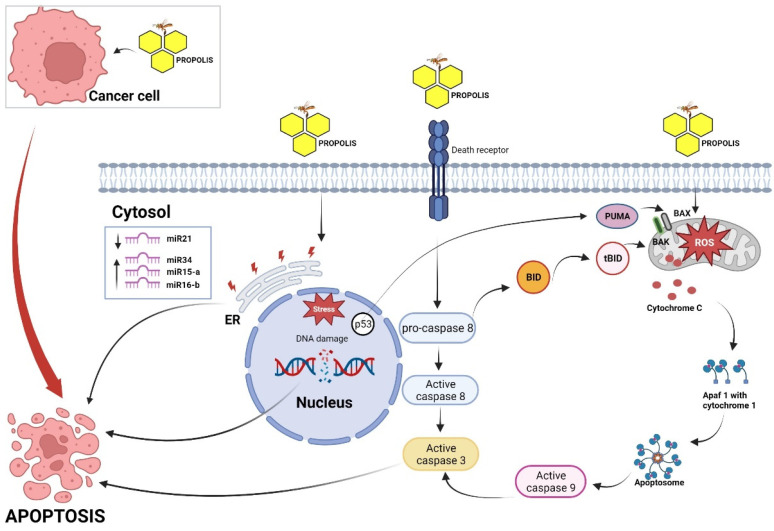
Proposed model illustrating the molecular mechanism of apoptosis induced by propolis in cancer cells. This figure was created by BioRender.com (accessed on 4 March 2023).

**Figure 4 pharmaceuticals-16-00450-f004:**
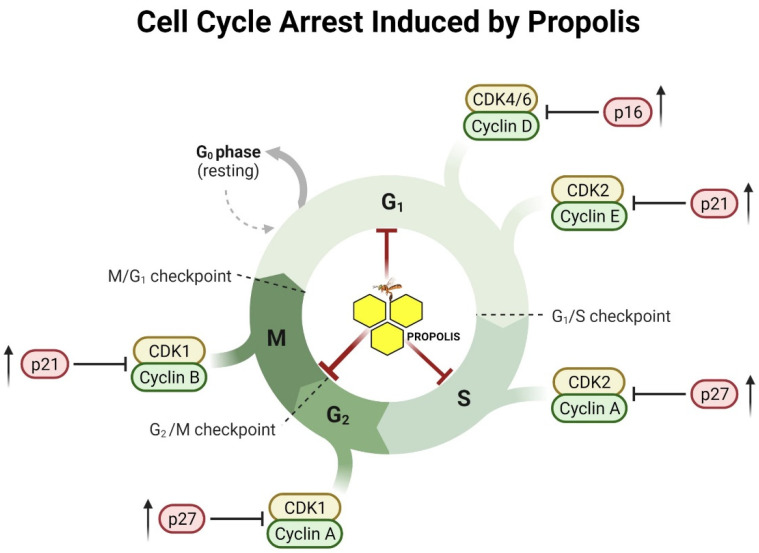
Proposed model illustrating the effect of propolis on cell proliferation and the cell cycle of cancer cells. This figure was created by BioRender.com (accessed on 4 March 2023).

**Figure 5 pharmaceuticals-16-00450-f005:**
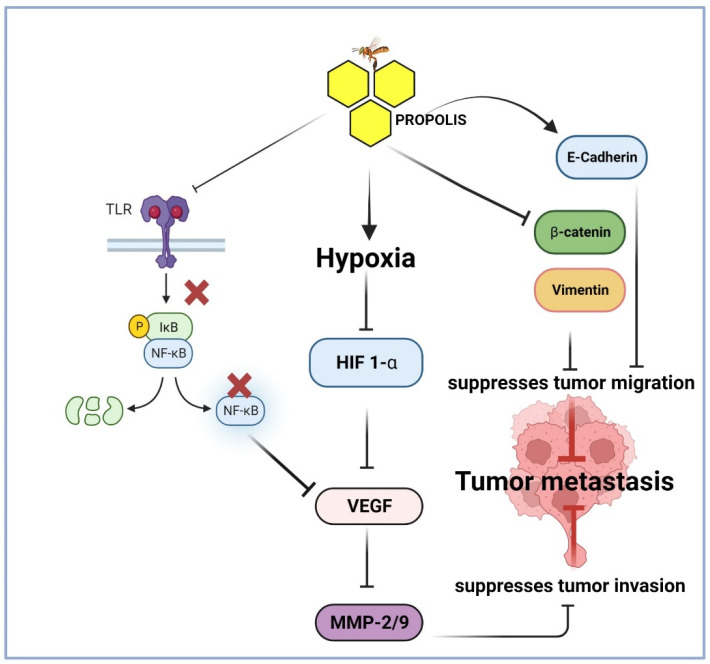
Proposed model illustrating the anti-metastatic molecular mechanism induced by propolis in cancer cells. This figure was created by BioRender.com (accessed on 4 March 2023).

**Table 1 pharmaceuticals-16-00450-t001:** Chemical composition of Propolis.

Chemical Groups Found in Propolis	Examples	Reference
Polyphenols	Flavonoids (e.g., quecentin, islapinin, chrysin, galangin, alnusin, pinocembrin, naringenin, pinostribon, ermanin, formononetin, and biochanin A)	[32,33,37,38,39,40,41,42]
	Phenolic esters (e.g., 4-phenyl coumarin)	
	Phenolic aldehydes (e.g., 4-hydroxybenzaldehyde)	
	Ketone derivatives (e.g., 3-oxo-oleanolic and 3-oxo-ursolic acids)	
	Flavanonols (e.g., dihydrokaempfero)	
Benzoic acid Derivatives	Gentisic acid, protocatechuic acid, salicylic acid phenylmethyl ester, and gallic acid	[34,35]
Benzaldehyde compounds	Vanillin, protocatechualdehyde, and caproic aldehydes	[34,35].
Cinnamic acid derivatives	Caffeic acid, isoferulic acid, cinnamic acid methyl ester, artepillin C, and capillartemisin A	[36,39,43]
Aliphatic hydrocarbons	Eicosine, tricosane, and heneicosane	[36,44]
Sugars	Fructose, glucose, glucitol, talose, ribose, and glycoside	[36,45]
Vitamins	B1, B6, C, and E	[36,45,46]
Amino acids	Alanine, cysteine, butyric acid, isoleucine, leucine, and valine	[36,46]
Fatty acids	Oleic, linoleic, stearic, eicosenoic, palmitoleic, and palmitic acid	[36,47]
Esters	Methyl palmitate, phenylethyl caffeate, benzyl benzoate, ethyl palmitate, tetradecyl caffeate, and stearic acid methyl ester	[36]
Alcohols	Benzyl alcohol, coumarine, and xanthorrhoeol	[36]
Minerals	Sodium, zinc, magnesium, potassium calcium, aluminium, and lead	[36,48]
Others	Enzymes, waxy acids, fatty acids, and aliphatic acids	[36]

**Table 2 pharmaceuticals-16-00450-t002:** Summary of Anticancer Mechanisms of Propolis on Human Cancer Cell lines.

Cancer Type	Cell Lines	Propolis Type/ Active Compound	Anticancer Mechanism(s)	Reference
Breast Cancer	MCF-7	CAPE	Modulation of the estrogen receptor	[169]
		New Zealand	Growth arrest of cells by downregulation of mortalin and activation of p53 tumor suppressor protein	[112]
		Indian propolis	Cytotoxicity “less dense and rounded cells”	[170]
		Brazilian propolis	Anti-proliferation	[171]
		Turkish propolis	Cell cycle arrest at G1 Apoptosis through increasing pro-apoptotic protein levels (p21, Bax, p53, p53-Ser46, p53-Ser15) and decreasing MMP Alters the miRNA expression of tumor suppression gene (miR 34, 15a, and 16-5p), miR 21 and breast cancer gene (BRCA ½)	[165]
	MDA-MB-231	Cuban red propolis	Apoptosis by modulation of PI3K/Akt, p38 MAPK, and ERK1/2 signaling pathways and enhancing ROS generation and loss of mitochondrial potential.	[130]
		CAPE	Inhibit invasion/metastasis and cell motility through blockage of voltage-gated sodium channels	[172]
		New Zealand propolis	Cell cycle arrest by downregulation of mortalin and activation of p53	[112]
		Serbian propolis	Cytotoxicity, proapoptotic, and antioxidative	[173]
	BT-474	Thai Apis mellifera propolis	Cell cycle arrest and apoptosis via p21 upregulation	[154]
Colon Cancer	HCT-116	Chinese propolis	Apoptosis	[137]
		Genistein	G2/M Cell cycle arrest and apoptosis	[152]
		Brazilian Propolis	Cell cycle arrest via Cip1/p21 expression	[153]
		Polish propolis	Apoptosis by DNA condensation	[124]
		Serbian propolis	Cytotoxicity, proapoptotic and antioxidative	[173]
	SW-480	Genistein	G2/M cell cycle arrest and apoptosis	[152]
	SW620	CAPE	Anti-angiogenesis and anti-metastasis via VEGF and MMPs inhibition	[24]
		Trigona incisa propolis	Apoptosis via increasing ROS	[125]
	HT-29	Cuban Propolis	Cell cycle arrest and apoptosis	[174]
		Indian propolis	Cytotoxicity “less dense and rounded cells”	[170]
	DLD-1	Propolis cinnamic acid	TRAIL-dependent apoptosis	[175]
		Brazilian propolis	Anti-proliferative	[171]
Prostate Cancer	PC-3	Kaempferol	Anti- proliferation by downregulation of PCNA and VCAM-1	[176]
		CAPE	Apoptosis by decreasing the cIAP-1/2	[129]
			Anti-inflammatory through suppression of COX-2/LOX and NF-κB signaling, 5-a reductase enzyme inhibition	[161]
	DU145, PC-3	Brazilian propolis	Downregulation of cyclin A, B, and D1, and Cdk and upregulation of p21	[97]
		Chrysin	Mitochondrial-mediated apoptosis and ER stress and down-regulating ERK1/2	[131]
		CAPE	PI3K/Akt downregulation	[132]
			Activation of the non-canonical Wnt-signaling pathway and synergistically act with docetaxel and paclitaxel	[177]
	LNCaP, DU145	Poland propolis	TRAIL-mediated apoptosis	[133]
Oral Cancers	SCC15/25, CAL27	CAPE	Cell cycle arrest p53 and Rb modulation Downregulation of oncogenes MIFT and K-Ras	[151]
	KB	Brazilian Brown propolis	COX-2 inhibition	[25]
	YD15, HSC-4 HN22	CAPE	Apoptosis via up-regulation of Bax and Puma proteins	[17]
	TW2.6	CAPE	Alteration of c-Jun N-terminal kinase, ERK1/2, NF-κB, and Akt signaling	[178]
	CAL27	Chrysin, Caffeic Acid, p-Coumaric Acid, and Ferulic Acid	PRODH/POX (proline degradation/proline oxidase) dependent apoptosis	[123]
Melanoma	A375	Chinese Propolis	Apoptosis, cell cycle arrest, and autophagy via NLRP1 Inflammatory Pathway	[145]
		Chrysin	Apoptosis (Bax activation) by upregulating p38 MAPK Downregulating the ERK1/2 signaling pathway	[128]
	A2058	CAPE	Suppression of the production of pro-inflammatory cytokines Production of anti-inflammatory cytokines (IL-10)	[161]
	B16F10	Galangin	Apoptosis via mitochondrial pathway and sustained activation of p38 MAPK	[127]
		Chrysin	Apoptosis and mitochondrial membrane potential loss through upregulating p38 mitogen-activated protein kinase (MAPK) and p62 Downregulation of tyrosinase activity (anti-melanogenesis) by modulating microphthalmia-associated transcription factor	[128]
Osteosarcoma	U2OS	New Zealand propolis	Cell cycle arrest by downregulation of mortalin and activation of p53	[112]
		Brazilian green propolis	Activation of p53 and growth arrest	[113]
			Apoptosis	[179]
Lung Carcinoma	A549	New Zealand propolis	Cell cycle arrest by downregulation of mortalin and activation of p53	[112]
		Brazilian green propolis	Activation of p53 and growth arrest	[113]
		Brazilian propolis	Anti-proliferative	[171]
Cervical Cancer	HeLa	Brazilian red propolis	Downregulation the expression alpha tubulin, tubulin, histone H3 and prostaglandin E synthase.	[180]
		New Zealand propolis	Cell cycle arrest mediated by downregulation of mortalin and activation of p53	[112]
Gastric Cancer	AGS	Iran propolis	Downregulation the mRNA expression of PLD1 gene	[120]
		Poland propolis	Interleukin (IL)-8 suppression	[181]
Pancreatic Cancer	PANC-1	Algerian propolis	Cell cycle arrest, Apoptosis, P-Glycoprotein Inhibition	[182]
		Vietnamese Trigona minor	Antiausterity	[183]
Oesophageal carcinoma	Eca9706, TE-1, and EC109	Galangin	Apoptosis and cell cycle arrest	[184]
Hepatocellular Carcinoma	HEp-2	Chinese and Egyptian propolis	Protection against doxorubicin-induced genotoxicity	[185]
Neuroblastoma	IMR32	New Zealand propolis	Cell cycle arrest by downregulation of mortalin and activation of p53	[112]
Fibrosarcoma	HT1080	Brazilian green propolis	Cell cycle arrest via activation of p53	[113]
Leukemia	CCRF-SB	Turkish propolis	Apoptosis	[186]

## Data Availability

Not applicable.

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
