# Peer review of "Propolis: A Detailed Insight of Its Anticancer Molecular Mechanisms"

_pharmaceuticals, 2023, doi:10.3390/ph16030450_

Round 1

Reviewer 1 Report

Authors have presented a review emphasizing “Propolis: A detailed Insights of its Anticancer Molecular Mechanisms”. Overall, the paper is well written however there are a few places where the unnecessary data is incorporated. However, these don't detract from the meaning of the text but a careful proofread could address these issues and improve the flow of the text. This manuscript could be a good addition to the literature related to anticancer activity of Propolis. In my opinion, the manuscript can be published in this journal, after the authors have addressed the following comments and questions:

·       Abstract section must be improved. This section not correlates with the whole study of the manuscript. This should be focused on anticancer activity of compound.

·       Authors are suggested to improve the introduction section by incorporating some content related to anticancer potential of propolis.

·       Also add some content in section “chemical constituents” related to bioavailability of various constituents of propolis.

·       There is no need of adding the section of different biological activities such as antibacterial, antiviral, antifungal and other…….Authors are suggested to remove these or compress these in a single paragraph so that theme/title of this review should be justified in terms of anticancer activity of propolis and mechanism.

·       Chemopreventive effect section is also not significant and not properly correlated. If possible remove this.

·       Figure 3 should be revised in a better way so that all the content will be clear to the readers.

·       The authors are suggested to check the typological and grammatical errors throughout the manuscript.

Author Response

Responses to reviewer #1:

Authors have presented a review emphasizing “Propolis: A detailed Insights of its Anticancer Molecular Mechanisms”. Overall, the paper is well written however there are a few places where the unnecessary data is incorporated. However, these don't detract from the meaning of the text but a careful proofread could address these issues and improve the flow of the text. This manuscript could be a good addition to the literature related to anticancer activity of Propolis. In my opinion, the manuscript can be published in this journal, after the authors have addressed the following comments and questions:

Authors’ response: We thank the reviewer for his positive remarks on our manuscript and for granting us the opportunity to share the updates on the anticancer molecular mechanisms of propolis with the scientific community.

Abstract section must be improved. This section not correlates with the whole study of the manuscript. This should be focused on anticancer activity of compound.

Authors’ response: We have improved the abstract section to correlate with the whole study of the manuscript focusing on the anticancer activity of propolis.

 Authors are suggested to improve the introduction section by incorporating some content related to anticancer potential of propolis

Authors’ response: We have improved the introduction section and incorporated some content to anticancer potential of propolis.

 Also add some content in section “chemical constituents” related to bioavailability of various constituents of propolis

Authors’ response: We have added some content related to the bioavailability of various constituents of propolis in section “Chemical Constituents, Bioavailability, and Biological Activities of Propolis”.

 There is no need of adding the section of different biological activities such as antibacterial, antiviral, antifungal and other…….Authors are suggested to remove these or compress these in a single paragraph so that theme/title of this review should be justified in terms of anticancer activity of propolis and mechanism.

Authors’ response: We have removed the section of different biological activities such as antibacterial, antiviral, antifungal and other and include only a single paragraph in section “Chemical Constituents, Bioavailability, and Biological Activities of Propolis” citing some reports on these biological activities to justify the theme/title of this review as suggested by the reviewer.

 Chemopreventive effect section is also not significant and not properly correlated. If possible remove this.

Authors’ response: We have removed the section “Chemopreventive effect of propolis” from the manuscript as per the reviewer’s recommendation.

Figure 3 should be revised in a better way so that all the content will be clear to the readers.

Authors’ response: To make the content in figure 3 clear to the readers, we have removed figure 3 and created three new figures (figure 3, 4, and 5) according to the molecular mechanisms of anticancer effect of propolis. Each figure is now placed following the related section in the text.

The authors are suggested to check the typological and grammatical errors throughout the manuscript.

Authors’ response: We have checked the typological and grammatical errors throughout the manuscript and made the necessary corrections as per the reviewer suggestion.

Reviewer 2 Report

The review ms by Altabbal and colleagues reports on the applications of Propolis, with focus on its anticancer effects. The manuscript is interesting and represent a nice contribution to the field.

Comments:

1- There are a few typos and grammar issues, the authors should go through the text fixing these little issues.

2- The last sentence of the Introduction is not understandable, please rewrite.

3- In section "Chemical Constituents of Propolis", I would suggest showing the data in another format. E.g. a table?. In the current state it is quite difficult to follow.

4- The section "Antibacterial Activities" needs checking to avoid conflicting information.

5- The section "Antibacterial Activities" is missing mechanistic data, the authors should explain the mechanisms a little further and not just list the data.

6- Please correct the scientific names under "Antifungal and antiprotozoa activity" section.

Author Response

Responses to reviewer #2:

The review ms by Altabbal and colleagues reports on the applications of Propolis, with focus on its anticancer effects. The manuscript is interesting and represent a nice contribution to the field.

Authors’ response: We thank the reviewer for his positive remarks on our manuscript and for granting us the opportunity to share the updates on the anticancer molecular mechanisms of propolis with the scientific community.

Comments:

1-There are a few typos and grammar issues, the authors should go through the text fixing these little issues.

Authors’ response: We have checked the typological and grammatical errors throughout the manuscript and made the necessary corrections as per the reviewer suggestion.

2- The last sentence of the Introduction is not understandable, please rewrite.

Authors’ response: We have re wrote this sentence to make it clear and understandable to readers.

3- In section "Chemical Constituents of Propolis", I would suggest showing the data in another format. E.g. a table?. In the current state it is quite difficult to follow.

Authors’ response: We have added a table showing the data of the "Chemical Constituents of Propolis" to make easy to follow for the readers.

4- The section "Antibacterial Activities" needs checking to avoid conflicting information.

Authors’ response: After careful revision of the whole manuscript, and to justify the theme/title of this review (focusing on the anti-cancer molecular mechanisms of propolis),  we have removed the section of different biological activities such as antibacterial, antiviral, antifungal and other and include only a single paragraph in section “Chemical Constituents, Bioavailability, and Biological Activities of Propolis” citing some reports on these biological activities. We believe that this option would most improve our paper. We hope that the reviewer will understand our wish to remove this detailed section and we appreciate your input on these changes. However, if the reviewer still suggests adding the "Antibacterial Activities", we will be happy to do.

5- The section "Antibacterial Activities" is missing mechanistic data, the authors should explain the mechanisms a little further and not just list the data.

Authors’ response: After careful revision of the whole manuscript, and to justify the theme/title of this review (focusing on the anti-cancer molecular mechanisms of propolis),  we have removed the section of different biological activities such as antibacterial, antiviral, antifungal and other and include only a single paragraph in section “Chemical Constituents, Bioavailability, and Biological Activities of Propolis” citing some reports on these biological activities. We believe that this option would most improve our paper. We hope that the reviewer will understand our wish to remove this detailed section and we appreciate your input on these changes. However, if the reviewer still suggests adding the "Antibacterial Activities", we will be happy to do.

6- Please correct the scientific names under "Antifungal and antiprotozoa activity" section.

Authors’ response: After careful revision of the whole manuscript, and to justify the theme/title of this review (focusing on the anti-cancer molecular mechanisms of propolis),  we have removed the section of different biological activities such as antibacterial, antiviral, antifungal and other and include only a single paragraph in section “Chemical Constituents, Bioavailability, and Biological Activities of Propolis” citing some reports on these biological activities. We believe that this option would most improve our paper. We hope that the reviewer will understand our wish to remove this detailed section and we appreciate your input on these changes. However, if the reviewer still suggests adding the "Antibacterial Activities", we will be happy to do.

Round 2

Reviewer 1 Report

Authors have significantly revised the manuscript in a better way thus it can be considered for publication.